# The Primary Microglial Leukodystrophies: A Review

**DOI:** 10.3390/ijms23116341

**Published:** 2022-06-06

**Authors:** Isidro Ferrer

**Affiliations:** Network Centre of Biomedical Research of Neurodegenerative Diseases (CIBERNED), Department of Pathology and Experimental Therapeutics, Bellvitge Biomedical Research Institute (IDIBELL), University of Barcelona, 08907 Barcelona, L’Hospitalet de Llobregat, Spain; 8082ifa@gmail.com; Tel.: +34-93-403-5808

**Keywords:** pigmented orthochromatic leukodystrophy, adult-onset orthochromatic leukodystrophy associated with pigmented macrophages, hereditary diffuse leukoencephalopathy with axonal spheroids, adult-onset (or dominant) leukodystrophy with axonal spheroids and pigmented glia, polycystic membranous lipomembranous osteodysplasia with sclerosing leukoencephalopathy, Nasu–Hakola disease, CRP, LKENP, CSFR1, AARS, TYROBP, TREM2

## Abstract

Primary microglial leukodystrophy or leukoencephalopathy are disorders in which a genetic defect linked to microglia causes cerebral white matter damage. Pigmented orthochromatic leukodystrophy, adult-onset orthochromatic leukodystrophy associated with pigmented macrophages, hereditary diffuse leukoencephalopathy with (axonal) spheroids, and adult-onset leukoencephalopathy with axonal spheroids and pigmented glia (ALSP) are different terms apparently used to designate the same disease. However, ALSP linked to dominantly inherited mutations in *CSF1R* (colony stimulating factor receptor 1) cause CSF-1R-related leukoencephalopathy (CRP). Yet, recessive ALSP with ovarian failure linked to *AARS2* (alanyl-transfer (t)RNA synthase 2) mutations (LKENP) is a mitochondrial disease and not a primary microglial leukoencephalopathy. Polycystic membranous lipomembranous osteodysplasia with sclerosing leukoencephalopathy (PLOSL; Nasu–Hakola disease: NHD) is a systemic disease affecting bones, cerebral white matter, selected grey nuclei, and adipose tissue The disease is caused by mutations of one of the two genes *TYROBP* or *TREM2*, identified as PLOSL1 and PLOSL2, respectively. TYROBP associates with receptors expressed in NK cells, B and T lymphocytes, dendritic cells, monocytes, macrophages, and microglia. *TREM2* encodes the protein TREM2 (triggering receptor expressed on myeloid cells 2), which forms a receptor signalling complex with TYROBP in macrophages and dendritic cells. Rather than pure microglial leukoencephalopathy, NHD can be considered a multisystemic “immunological” disease.

## 1. Introduction

Leukodystrophies are genetic diseases primarily affecting the production, processing, or development of myelin in the central nervous system (CNS) and components of the white matter, such as oligodendrocytes and astrocytes. The broader term leukoencephalopathy refers to all white matter diseases, either hereditary or acquired, which can be due to neuronal or systemic pathologies.

Most leukodystrophies appear in children. Adult leukodystrophies are less common and subject to continuous categorization. Adult leukodystrophies and leukoencephalopathies can be categorized into hypomyelinating leukodystrophies, demyelinating leukodystrophies, leukodystrophies considered primary astrogliopathies, leukodystrophies considered primary microgliopathies, miscellaneous leukodystrophies, inherited vascular encephalopathies, and inherited leukodystrophies of unknown origin [1,2,3,4,5,6]. The present review is focused on a particular group of adult leukodystrophies in which the cause of the disease is linked to mutations in genes expressed in microglial cells. For this reason, these leukodystrophies are considered primary microglial leukodystrophies or primary microglial leukoencephalopathies [1,7,8,9]. The discovery of new mutations in various genes has allowed the modeling of new concepts of the microglial leukoencephalopathies. The present review aims to update this topic covering clinical, radiological, neuropathological, genetic, and pathogenetic aspects of these disorders.

## 2. Adult-Onset Leukodystrophy with Axonal Spheroids and Pigmented Glia (POLD, HDLS, ALSP): CRP Linked to Mutations in *CSF1R*, and LKENP (Leukoencephalopathy, Progressive with Ovarian Failure) Linked to Heterozygous Mutations in *AARS2*

Pigmented orthochromatic leukodystrophy (POLD) or adult-onset orthochromatic leukodystrophy associated with pigmented macrophages was described by van Bogaert and Nyssen in 1936 [10]; several sporadic and familial cases have been reported to date [11,12,13,14,15]. Hereditary diffuse leukoencephalopathy with (axonal) spheroids (HDLS) was reported in 1984 [16] and has been documented as new sporadic and familial cases [17,18,19,20]. The similarities between POLD and HDLS suggest that these diseases represent a single entity [21]. The term adult-onset (or dominant) leukodystrophy (or leukoencephalopathy) with axonal spheroids and pigmented glia (ALSP) was proposed to describe similar conditions [22,23,24,25,26,27]. More recently, ALSP was shown to encompass POLD and HDLS, and isconsidered the same entity [28,29,30,31].

Genetic studies have shown that at least three different diseases were designated with the names POLD/HDLS/ALSP.

Dominantly inherited mutations in colony stimulating factor 1 receptor (*CSF1R*) cause CSF-1R-related leukoencephalopathy [32] (ALSP-CSFR1: CRL, OMIM # 221820). Recessive mutations in alanyl-transfer (t)RNA synthase 2 (*AARS2*) cause leukoencephalopathy, progressive with ovarian failure [33] (ALSP-AARS2: LKENP, OMIM # 615889). A third, still poorly defined group of POLD-like leukoencephalopathies with or without epilepsy, is linked to mutations in *AARS1* [34,35].

### 2.1. Clinical Features

#### 2.1.1. ALSP-CSFR1 or CRL

A large study comprising 90 families with the *CSFR1* mutation revealed the mean age at onset to be 43 years (range 18–78 years), mean age at death 53 years (range 23–84 years), with a mean disease of duration 6.8 years (range 1–29 years); clinical symptoms in women appear at a younger age [36].

Behavioural and personality changes, memory impairment, executive dysfunction, and dementia are common symptoms. These are often accompanied by motor dysfunction including pyramidal signs (spasticity, hyperreflexia, hemiparesis, quadriparesis), sensory deficits, depression, and parkinsonian symptoms and signs such as gait impairment, rigidity, bradykinesia, postural instability, and tremor [26,27,36,37,38,39,40,41,42,43,44]. Speech difficulty, non-fluent aphasia, and symptoms mimicking frontotemporal dementia behavioural variant (FTD-bv) have been described [31,45,46,47]. Less common manifestations are corticobasal syndrome and stroke [40]. Cerebellar and bulbar/pseudobulbar signs may occur at advanced stages, whereas seizures may appear at the onset of the disease. Peripheral neuropathy has been noted in some patients [46,48]. Yet, whether peripheral neuropathy belongs to CSF1R-related spectrum is an open question requiring further investigation [46]. Involvement of the optic nerve is very rare [49].

Considering the variability of clinical symptoms, misdiagnoses such as frontotemporal dementia, parkinsonism, multiple sclerosis, normal-pressure hydrocephalus, and vasculitis of the central nervous system, among others, are commonplace [50,51,52]. Differential diagnosis covers several entities, mainly adult-onset leukodystrophies [2,31].

#### 2.1.2. ALSP-AARS2 or LKENP

Dallabona et al. [33] described another group of patients with a progressive leukoencephalopathy starting in childhood or in early adulthood, characterized by progressive gait ataxia, tremor, spasticity, dystonia, dysarthria, and cognitive decline. The affected women presented with ovarian failure and amenorrhea, followed by neurodegeneration, or presented ovarian failure during the progression of the neurological disorder. The investigation of mitochondrial function in two patients identified cytochrome c oxidase deficiency. Ragged-red fibres were absent. None of the patients had signs of cardiomyopathy. Brain MRI showed leukoencephalopathy with the involvement of left–right connections, descending tracts, and cerebellar atrophy. Next-generation sequencing revealed compound heterozygous mutations in *AARS2*, which encodes mitochondrial alanyl-tRNA synthetase, in both patients. Functional studies in yeast confirmed the pathogenicity of the mutations in one patient [33]. Additional cases of leukoencephalopathy linked to *AARS2* mutations have been reported [53].

#### 2.1.3. Non-ALSP AARS1 Mutations

Autosomal-recessive *AARS1* mutations were described in three individuals (two siblings and an unrelated individual) affected by severe infantile epileptic encephalopathy with a central myelin defect and peripheral neuropathy [34]. Developmental and epileptic encephalopathy 29 (DEE29; OMIM# 616239), or epileptic encephalopathy, early infantile, 29 (EIEE29) is the term used to designate this disease. Recently, eleven individuals bearing *AARS1* mutations with two different phenotypes were described. One group of patients suffered early infantile-onset disease, resembling DEE29. The second group had a later-onset disorder, and the clinical symptoms were not homogeneous, having in common cognitive impairment, regression of mobility, spasticity, and tetraparesis; dystonia and epilepsy were present in two patients. The most characteristic feature was a progressive posterior predominant leukoencephalopathy evolving to include the frontal white matter. AlaRS activity was significantly reduced in five affected individuals with both early infantile-onset and late-onset phenotypes [35].

### 2.2. Radiological Findings

Atrophy and increased signal in the white matter, thinning of the corpus callosum, abnormal signalling of the pyramidal tracts, and enlarged ventricles are seen on MRI. The subcortical and periventricular white matter shows T2-weighted and FLAIR hyper-intensities with no gadolinium-enhanced lesions; these hyperintensities are usually diffuse but they can be confluent, with asymmetric distribution. Calcifications in the white matter are found in about one half of cases; calcifications in the frontal periventricular white matter are common in cases linked to *CSFR1* mutations (CRL), but they are absent in cases linked to *AARS2* mutations (LKENP) [24,26,33,36,38,39,41,44,54,55,56,57,58,59,60].

In CT images, calcifications mainly involve the frontal white matter adjacent to the anterior horns of the lateral ventricles and the parietal subcortical white matter; calcifications have a symmetrical, “stepping stone” appearance in the pericallosal regions [61]. Thin-slice CT techniques are necessary to detect small calcifications [41]. The cerebellum and the brain stem appear normal.

MR spectroscopy shows changes in metabolite concentrations not only in patients with HDLS linked to *CSFR1* mutation and also in asymptomatic *CSF1R* mutation carriers [62].

Differential diagnosis of leukoencephalopathies with calcifications includes COL4A1-related disorders and primary familial brain calcifications with leukoencephalopathy [59].

Leukoencephalopathy linked to *AARS1* mutations is characterized on MRI by progressive posterior predominant leukoencephalopathy evolving to include the frontal white matter [35].

### 2.3. Neuropathology

The characteristic lesions are: (a) bilateral and confluent demyelination of the cerebral white matter sparing the cortico-subcortical U fibres, and involvement of the corpus callosum and internal capsule; (b) axonal damage with spheroids in the white matter; (c) infiltration of macrophages filled with neutral lipids; and (d) cytoplasmic deposits in macrophages, astrocytes, and oligodendrocytes stained with periodic acid–Schiff, Sudan black, and Klüver–Barrera that are autofluorescent in paraffin sections [11,12,38,39,40,41,45]. Axonal damage is best seen with silver-based methods and immunohistochemistry with antibodies against neurofilaments, β-amyloid precursor protein, and ubiquitin. Electron microscopy reveals that spheroids are mainly composed of neurofilaments; less frequently, they also contain neurotubules, mitochondria, and altered mitochondria, together with granular material [20,63]. Pigments contain electron-dense granular material with lamellar or fingerprint arrangement reminiscent of ceroid-lipofuscin [11,12,20,25] (Figure 1 and Figure 2). Characteristic lesions are also seen in cerebral biopsy samples.

The abundance of spheroids and pigmented glia varies from one case to another and with disease progression [25,37,64,65,66]. White matter lesions have been categorized in three stages: (1) white matter with numerous spheroids in a background of well-myelinated fibres, (2) moderate loss of myelinated fibres with a sparse-to-moderate number of spheroids, and (3) leukodystrophy pattern of confluent axonal and myelin loss [67]. This pattern is in line with pioneering observations noting that: (i) the formation of spheroids is an early event in ALSP, (ii) spheroids are more abundant in areas of partial demyelination than in areas of extensive demyelination, and (iii) the loss of oligodendrocytes occurs in regions of extensive demyelination but not partial demyelination [45]. Similar lesion-based stages have been proposed, with two additional conclusions: (i) shape, density, and subsets of microglia change with stage progression, and (ii) an increase in IBA-1, CD-68, CD-163-, and CD-204-immunoreactive cells precedes the loss of axons [66]. Neuropathological features were reported in a non-affected *CSF1R*-carrier who died of tuberculosis. Patchy demyelination and axonal loss predominate in the subcortical white matter, whereas pigmented microglial cells are distributed throughout the white matter preceding myelin and axonal loss [18]. Interestingly, activated microglia are spatially restricted, rather than diffusely distributed, in ADLS cases bearing *CSF1R* mutations [41]. Quantitative studies show a predominance of axonal spheroids and axonal depletion in areas with almost complete absence of ramified microglia, thus suggesting that loss of microglial ramification, indicative of activation, precedes axonal spheroid formation [67].

The caudate, putamen, thalamus, hypothalamus, hippocampus, substantia nigra, nuclei of the brain stem, and cerebellar cortex are unaffected or only very mildly affected [31].

Axonal swellings have been noted in skin nerves [20].

Early biochemical studies in the white matter revealed normal levels of galactolipids, decreased polyunsaturated fatty acids, and increased plasmalogens [11]. No genetic determination was available at that time.

No neuropathological studies are available in cases with leukoencephalopathy linked to *AARS1* mutations.

### 2.4. Genetics

#### 2.4.1. CRL

Heterozygous mutations in the colony stimulating factor 1 receptor gene (*CSF1R*) are causative of HDLS [32]. This seminal observation was followed by several studies showing *CSF1R* mutations in both familial and sporadic cases of HDLS [27,36,40,41,42,49,51,58,68,69,70,71,72,73], POLD [29], and ALSP [36,38,41,44,46,74,75,76,77,78]. Most mutations are located in the tyrosine kinase domain (TKD), but others are located outside the region encoding TKD [36,79]. No correlation is apparent between phenotype and genotype, considering patients with mutations in proximal kinase domain compared with the distal kinase domain, except for seizures, which are more frequent in patients who had mutations in the proximal kinase domain [36]. Individuals carrying truncating mutations or mutations that trigger nonsense-mediated mRNA decay have an earlier disease onset [80]. Intrafamilial heterogeneity is common in some families [73]. However, about 40% of cases have no familial history; the age-dependent penetrance may cause an overestimation of such cases [36].

Homozygous mutations in *CSFR1* cause a paediatric-onset leukodystrophy accompanied by brain malformations (absence of the corpus callosum), and lack of microglia, presenting clinically with delayed development and epilepsy [81]. Bi-allelic *CSF1R* mutations have also been reported in seven affected individuals from three unrelated families who had, in addition to early onset HDLS-like neurological disorders, brain malformations and skeletal dysplasia compatible with dysosteosclerosis or Pyle disease [82]. These distinct disorders do not belong to the group of leukoencephalopathies discussed in this review.

#### 2.4.2. LKENP

*AARS2* mutations cause a recessive form of ALSP now named leukoencephalopathy, progressive with ovarian failure (LKENP). This was identified in six patients with childhood- to adulthood-onset signs of ataxia, spasticity, and cognitive decline, leukoencephalopathy, involvement along the corticospinal tract, and cerebellar atrophy; all female patients had ovarian failure [33]. Later, biallelic mutations in *AARS2* were reported in five patients from four different families presenting in adulthood with cognitive impairment, neuropsychiatric symptoms, and upper motor neuron signs; a cerebral biopsy in one case demonstrated typical pathological features and females showed early ovarian dysfunction [50]. Another young woman bearing a new *AARS2* mutation, with similar clinical, radiological, and pathological features on a cerebral biopsy, did not show ovarian abnormalities [83]. New cases of leukodystrophy linked to *AARS2* mutations have been reported in males and females with variable clinical course; some are accompanied by peripheral neuropathy [53,84,85,86,87]. An *AARS* variant has been proposed as the likely cause of Swedish-type hereditary leukoencephalopathy with spheroids [88].

The clinical spectrum linked to *AARS2* mutations includes a rare case with retinopathy and optic atrophy in an 18-month-old Korean boy with bilateral optic atrophy, peripheral retinal bone spicule pigmentation, absent patellar reflexes, and demyelinating polyneuropathy, together with cerebellar and supratentorial white matter changes with areas of restricted diffusion, and dorsal column signal abnormalities on MRI [89]. Another case bearing an *AARS2* mutation suffered from ataxia without leukoencephalopathy [90]. Ragged-red fibres were observed in the striated muscle in one case with *AARS2*-related leukoencephalopathy; gradual improvement in motor function was observed with intravenous coenzyme complex treatment [91]. These variegated disorders linked to *AARS2* mutations, other than LKENP, are distinct diseases, which are the focus of the present review.

Finally, exome sequencing identified mitochondrial alanyl-transfer tRNA synthetase (*AARS*) mutations in unrelated infantile mitochondrial cardiomyopathy and multiple oxidative phosphorylation defects [92,93]. Histological evaluation of muscular biopsies in infantile mitochondrial cardiomyopathy and multiple oxidative phosphorylation defects reveals massive mitochondrial accumulation and cytochrome c oxidase-negative fibres [93]. Biochemical studies show decreased activity of mitochondrial respiratory chain complexes I and IV with a mild decrease in complex III activity in skeletal and cardiac muscle in infant patients with *AARS* mutations [93,94]. The disease is referred as combined oxidative phosphorylation deficiency 8 (COXPD8, OMIM#614096). This disease is not a leukodystrophy and it is not discussed here.

#### 2.4.3. Non-ALSP AARS1 Mutations

Mutations in alanyl-tRNA synthetase 1 (AARS1) may show two distinct phenotypes with neurological symptoms: (i): epileptic encephalopathy with deficient myelination (DEE29; OMIM# 616239) [34,35]; and (ii): progressive posterior predominant leukoencephalopathy [34,35].

*AARS1* gene mutations have also been reported in Charcot–Marie–Tooth type 2N [95], and in association with recurrent hepatic failure [96].

### 2.5. Pathogenesis

CSFR1 is a transmembrane tyrosine kinase receptor expressed on the surface of mononuclear phagocytes and microglia [97,98]. Upon activation after binding with its ligand colony stimulating factor-1 (CSF-1) and interleukin-34, CSFR1 autophosphorylates and initiates signal transduction in microglia; balanced CSF-1/CSF-2 receptor signalling is required for microglial homeostasis [99]. Flow cytometry analysis demonstrates the altered expression of antigen presentation-related and migration-related molecules, altered cytokine production, and abnormal phagocytic activity in monocytes from ALSP cases [100]. Reduced lower band microtubule-associated protein 1 light chain 3-II (LC3-II), which is activated in autophagy, is observed in mutant cells when compared with wild-type cells, further suggesting impaired autophagy linked to *CSFR1* mutation [46].

Mutations in the *CSF1R* homologues in zebrafish, csf1ra and csf1rb, produce aberrant microglia density and regional loss of microglia [101]. The inactivation of one Csf1r allele is sufficient to cause ALSP-like disease in mice; Csfr1 haplo-insufficient mice show lateral ventricle enlargement, thinning of the corpus callosum, dysmyelinated axons and axonal spheroids in the white matter, and increased expression of cytokines consistent with microglial activation [99].

These observations point to the primary pathogenic role of microglia in the pathogenesis of a subset of leukodystrophies (and impaired development of the nervous system) linked to *CSFR1* mutations, which is in accordance with the roles of microglia in the normal brain and in neurological diseases [102,103]. Based on these findings, *CSF1R*-related leukoencephalopathy (CRP) may be considered representative of primary microgliopathies [8]. However, AARS2 encodes a mitochondrial class-II aminoacyl-tRNA synthetase that participates in mRNA translation, specifically amino-acylating alanyl-tRNA. Therefore *AARS2*-related leukoencephalopathy, progressive with ovarian failure (LKENP) is a mitochondrial leukoencephalopathy. The rare group of leukoencephalopathies linked to mutations in *AARS1* are not microgliopathies.

### 2.6. Phenotype Differences between CRP and LKENP

Patients with CRP and LKENP mutations have similar clinical symptoms, radiological alterations, and leukodystrophy with axonal spheroids and pigmented glial cells, but they also each have particular characteristics: the age of onset is earlier in patients with LKENP (mean 26 years) when compared with CRP (mean 42 years); white matter atrophy and corpus callosum thinning is more marked in CRP; ventricular abnormalities and calcifications in the frontal periventricular white matter are not seen in LKENP; and white matter rarefaction, which is suppressed on fluid-attenuated inversion recovery MRI sequences, is found in LKENP but not in CRP [104]. Ovarian failure occurs in the majority of female LKENP [14,50,86]. However, premature ovarian failure preceding neurological symptoms is only rarely noted in CRP [105].

POLD/HDLS/ALSP is probably a mis-/underdiagnosed entity [28,106]. In the majority of cases, the diagnosis is made at autopsy, or at cerebral biopsy [20,27,46,52,58,63,83,107,108] (Figure 3). Moreover, most cases reported in the pre-genetic era cannot be properly categorized by current criteria [3].

**Figure 3 ijms-23-06341-f003:**
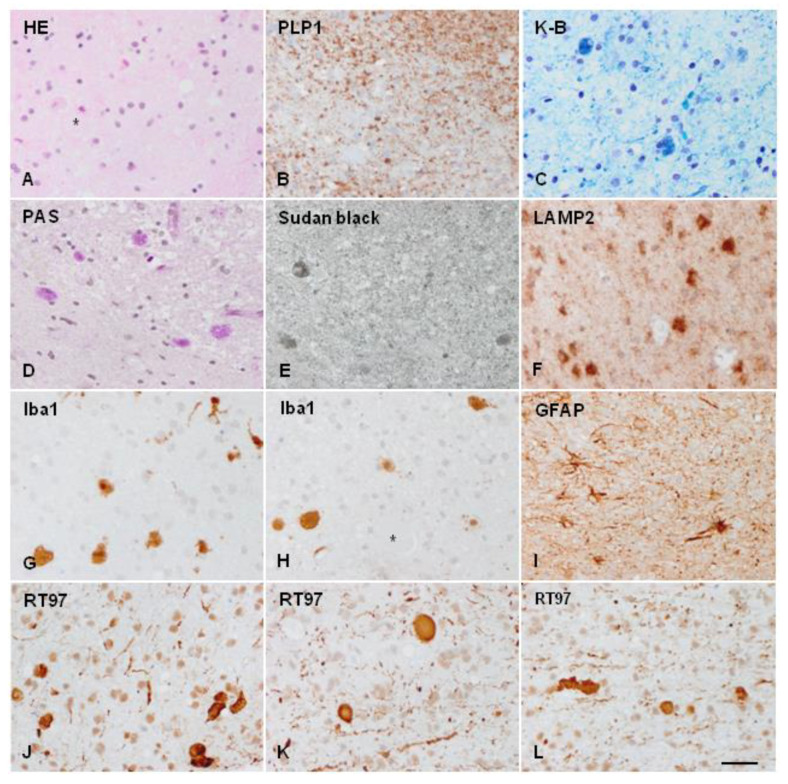
ALSP cerebral biopsy sample. (**A**) Axonal ballooning (asterisks) as seen in haematoxylin and eosin-stained sections. (**B**) Reduced proteolipid protein 1 (PLP1, a protein component of myelin) immunoreactivity. (**C**) Myelin loss and deposits of pigment in glial cells, as revealed by Klüver–Barrera (K-B) staining, demonstrating the presence of phospholipids in the pigment. (**D**) Pigmented glial cells are stained with PAS, and (**E**) Sudan black, showing the deposits of glucolipids and complex lipids; (**F**) glial cells showing the accumulation of lysosome-associated membrane glycoprotein 2 (LAMP2), a marker of autophagy. (**G**,**H**) Globose microglial cells and macrophages stained with allograft inflammatory factor 1(Iba1) antibody. (**I**) Moderate astrogliosis as revealed with glial fibrillary acidic protein (GFAP) immunohistochemistry. (**J**–**L**) different fields showing nerve fibre loss and abundance of axonal ballooning as revealed with the anti-neurofilament (200 kDa) antibody RT97. Paraffin sections, bar = 50 μm. Details of case report reference [109,110].

## 3. Polycystic Membranous Lipomembranous Osteodysplasia with Sclerosing Leukoencephalopathy (PLOSL; Nasu–Hakola Disease)

PLOSL or membranous lipodystrophy or Nasu–Hakola disease (NHD), first described in the early seventies in Japan and Finland [111,112,113,114], has been reported in several countries worldwide. PLOS is inherited in an autosomal recessive manner with no gender preference. The disease can be caused by mutations in two different genes, either TYROBP or TREM2, whose mutations cause slightly different versions of the disease designated as PLOSL1 (OMIM # 221770) and PLOSL2 (OMIM # 618193), respectively.

### 3.1. Clinical Features

The disease usually manifests in the third decade by multiple cyst-like lesions and loss of bone trabeculae, causing pain and tenderness, mainly in feet and ankles; fractures following minor accidents are common. The osseous stage is followed by a progressive neurological disease consisting of abnormal personality and behaviour, episodes of depression and euphoria, loss of social inhibition, memory deficits, and apathy, together with spasticity, postural dyspraxia, and involuntary movements. Urinary incontinence, loss of libido, and sexual impotence are early symptoms in some cases. Seizures may be present. Patients die in a vegetative state before the age of 50 years [111,112,115,116,117,118,119,120,121,122,123,124,125]. Due to the first appearance of osseous lesions in most cases, neuropsychological tests and functional neuroimaging can be useful in identifying subclinical bone alterations in the early stages of NHD [126,127]. Progressive spastic paraplegia and dementia without skeletal symptoms, but accompanied by polycystic osteodysplasia following X-ray examination, are clinical manifestations in some subjects [126]. In some cases, skeletal symptoms are not apparent, and patients complain of behavioural and neurological deficits. NHD cases without apparent skeletal symptoms occur in patients with mutations in *TREM2*, but not *TYROBP* [128,129,130,131,132]. Osseous pathology is not always present or diagnosed [121,125,133]. Radiological examination of bones is recommended in patients with neurological deficits clinically suggestive of NHD.

### 3.2. Radiological Examination

In NHD, cyst-like lesions are typically found in the carpal and tarsal bones and in the fingers [115,121,134].

Brain CT scans show cerebral atrophy with enlargement of the frontal regions of the lateral ventricles and calcifications in the globus pallidus. MRI reveals diffuse high density in the cerebral white matter and reduced density in the basal nuclei on T2-weighed images; white matter lesions extend from the periventricular areas to the periphery, sparing the cortico-cortical arcuate fibres [113,120,121,135,136,137,138,139]. Hypoperfusion of the grey matter has been noted on SPECT and PET studies [140,141].

### 3.3. General Pathology

Bone lesions are membrane-cystic lesions that destroy the osseous tissue. Membranes are convoluted, eosinophilic, PAS-positive, and autofluorescent; they contain carbohydrates, phospholipids, fatty acid crystals, hydroxyapatite crystals, and collagen. Under electron microscopy, the membranes mainly form tubular and saccular structures, and contain granular material. Cysts are filled with triglycerides. In addition, the neighbouring blood vessels show reduced lumen, enlargement of the inner elastic membrane, and degeneration of the muscularis layer [114,120,142,143,144,145]. Examination of a transiliac bone biopsy sample from one patient revealed disordered lamellar collagen fibril arrangement and increased matrix mineralization, pointing toward the involvement of osteoclast defects in this condition [146].

In addition to the bones, membranous lipodystrophy appears as a form of degeneration of the adipose tissue [144]. Several tissues are also affected, such as the cutaneous and perivisceral adipose tissue, bone marrow, alveolar septa of the lungs, and rectal mucosa [120,126,147].

### 3.4. Neuropathology

The central nervous system in NHD is decreased in size and has a reduced consistency of the cerebral white matter, atrophy of the frontal and temporal lobes, and dilated ventricles. The globus pallidus presents with brownish discoloration on macroscopic examination. Histological sections processed for myelin staining exhibit generalized symmetrical demyelination of the cerebral white matter with preservation of the arcuate subcortical fibres, corpus callosum, and white matter of the cerebellum. There is marked astrogliosis, but with the discrete presence of macrophages filled with sudanophilic material, and the absence of inflammatory infiltration. Axonal degeneration accompanied by axonal spheroids is common. Axonal spheroids contain neurofilaments, mitochondria, vesicles, and granular material [125,126,148,149]. The blood vessels in the white matter have increased thickness and multi-layering of the basement membrane [120,150].

The grey matter has less notable pathology [149]. The globus pallidus shows variable neuron loss and astrocytic gliosis, and numerous calcospherites containing calcium, iron, and glycoprotein matrix [126,139,148,149,151]. The thalamus and hippocampus show discrete neuronal loss in most cases. Marked neuronal loss and astrocytic gliosis in the caudate nucleus, putamen, substantia nigra, and thalamus (particularly in the dorsomedial and anterior nucleus) is found in some patients [149]. Severe thalamic degeneration occurs only rarely [152].

Alzheimer’s disease pathology (neurofibrillary tangles and senile plaques) is as common in NHD as in the control population [153]. However, a recent report described a 51-year-old female with NHD, bearing a homozygous mutation (Q33X) of TREM2 gene, and showing spots of neurofibrillary tangle pathology in the neocortex, but sparing the mesial temporal structures on post-mortem examination [154].

The percentage of short carbon chain non-hydroxyl fatty acids of sulfatide and the percentage of palmitic acid of ganglioside in the cortex in NHD were both increased in comparison to controls [155].

Peripheral neuropathy with axonal and segmental degeneration has been reported in a few cases [126,156].

### 3.5. Genetics

PLOSL is caused by mutations in either of two genes, *TYROBP (DAP12*) and *TREM2* [122,131,141,157,158,159,160,161,162]. The first case of NHD had a mutation in *DAP12* [122]. Mutations in these genes produce two variants: PLOSL1 (OMIM # 221770) and PLOSL2 (OMIM # 618193), respectively.

*TYROBP* encodes a tyrosine kinase binding protein (TYROBP) or DNAX-activating protein 12 (DAP12), or killer cell activating receptor-associated protein (KARAP), which is a membrane receptor component expressed in NK cells, B and T lymphocytes, dendritic cells, monocytes, macrophages, and microglia [162,163,164,165,166,167]. The complex adaptor protein KARAP/DAP12/TYROBP associates with multiple immunoglobulin domains and C-type lectin [168].

*TREM2* encodes the triggering receptor expressed on myeloid cells 2 (TREM2), which forms a signalling complex with TYROBP in macrophages and dendritic cells [157,169]. TREM2 and TYROP interact functionally to form a receptor signalling complex. However, it should be emphasized that DAP12 is a signalling adaptor that associates not only with TREM2, but also with multiple immunoglobulin domains and C-type lectin receptors. The disruption of TYROBP/TREM2 signalling in Nasu–Hakola disease leads to enhanced activation of their downstream effector Syk in various brain regions [170].

*TREM2* mutations have also been reported in cases of frontotemporal dementia without bone cysts [128,129,132].

### 3.6. Pathogenesis

Osteoclast differentiation is markedly altered in TREM2- and DAP12-deficient PLOSL patients, resulting in impaired bone resorption capacity [170,171]. This observation suggests a key role for TREM2 and DAP12 in the differentiation of myeloid precursors into functional multinucleated osteoclasts [171,172]. DAP12-deficient mice develop osteopetrosis; osteoclasts induced from DAP12−/− bone marrow cells produce immature cells with impaired bone resorption activity [173]. Studies using TREM2 RNA interference in pre-osteoclasts reported the same conclusion [174]. KARAP/DAP12-deficient mice also demonstrate that the complex specifically controls osteoclast differentiation [175].

In the nervous system, TREM2 and DAP12 are co-expressed in microglia, and TREM2/DAP12-positive cells are in close apposition with CNP+ oligodendrocytes during development [176]. This complex is involved in immune responses in microglia, macrophages, and dendritic cells in the brain [177,178]. TREM2 also promotes a putative neuroprotective microglia phenotype [179]. KARAP/DAP12-deficient mice demonstrate that the complex controls microglial differentiation [174]. TREM2 deficiency impairs chemotaxis and microglial responses to neuronal injury [180].

Microglia derived from human stem cells carrying missense TREM2 mutations do not express cell surface TREM2, but they show normal functional responses to TLR4 stimulation and normal phagocytic uptake of bacteria and acetylated LDL [181].

Microglial cells in NHD show reduced DAP12 immunoreactivity but preserved IBA1 expression; CD33-immunoreactive microglia counts are not altered in NHD when compared with control cases [182,183]. However, individual variability has also been reported [184]. DAP12 expression was found in numerous microglia in one NHD case with a homozygous DAP12 single-base substitution. In contrast, levels of both DAP12 and TREM2 were much lower in three other cases; mild activation of microglia in the cerebral white matter as well as little expression or none of DAP12 occurred in these three cases. These observations reveal the variable expression of the DAP12 and TREM2 genes in NHD, probably depending on the characteristics of the mutation [184].

Other studies have shown altered capacities for oxidative stress in microglia in NHD. Thus, gp91phox, one of the catalytic subunits of NAD, is overexpressed in microglia in NHD [185]. Induced microglial cells developed from peripheral blood from an NHD patient express delayed but stronger inflammatory responses compared with those from healthy controls [186]. Together, these observations indicate abnormal microglia function in NHD.

However, the implication of altered microglia in the pathogenesis of NHD is not completely understood. DAP12-deficient mice have immature oligodendrocytes arrested in the vicinity of the thalamus. This is accompanied by reduced local myelin formation and thalamic hypomyelinosis with synaptic degeneration in these mice [173].

The loss of TREM2 in transgenic mice results in a decrease in expression of oligodendroglia- and myelin-related genes [187]. However, the number of cells expressing G protein-coupled receptor 17 (GPR17), an intrinsic timer of oligodendrocyte differentiation and myelination, does not differ in control and NHD brains. These findings suggest that GPR17-positive pre-oligodendrocytes do not play a role in the development of leukoencephalopathy [188]. LC3 is increased in oligodendrocytes, pointing to the involvement of altered autophagy in oligodendrocytes, at some stage, in the white matter lesions in PLOSL [189].

DAP12-deficient mutants also have synaptic degeneration and impaired synaptic function [173]. Complex synaptic deficits have been demonstrated as well in KARAP/DAP12-deficient mice [190]. Since KARAP/DAP12 is expressed in microglia, it has been suggested that impaired synaptic activity in these mice is secondary to altered microglia/neuron interaction [190].

TREM2 is required for synapse elimination and normal brain connectivity [191]. The loss of TREM2 in transgenic mice produces a decrease in synaptic protein levels, most markedly in the hippocampus, together with reduced synaptic plasticity in adulthood [187].

It has been suggested that the loss of DAP12/TREM2 function in microglia might not be primarily responsible for the neuropathological phenotype of NHD [169]. However, we do not have sufficient evidence for the sustained direct cause–effect relationship between microglial alterations resulting from DAP12 and TREM2 mutations, and the altered myelination of the cerebral white matter, loss of synapses, reduced numbers of neurons in selected regions, such as the thalamus, and dramatic calcium and pigment deposition in the globus pallidus.

Indirect data provide some insight into alternative pathways that are activated in NHD. Syk, a participant in the DAP12 receptor signalling pathway, is hyperphosphorylated in neurons in NHD independently of the degree of P-Syk expression in microglial cells, which does not differ between NHD cases and controls [170]. Alternative pathways might be dysfunctional in NHD, which would explain the connection between microglial alterations and neuronal damage. CX3CL1-CX3CR1 signalling is an important communication pathway between neurons and microglia [192].

TREM2 research in the CNS is largely focused on microglia, and the signalling pathways in these cells are well established in validated in vitro and in vivo studies. DAP12 has activating and inhibitory functions in white blood cells; the fate of any particular effect is probably dependent on its interactions with a large number of ligands and adaptors. Besides the functional activity in NK cells that implies the activation of SRC kinases, and the subsequent phosphorylation of tyrosine residues in the immunoreceptor tyrosine-based activation motif of DAP12, other models have been proposed to explain the effects of DAP12 on the activation and inhibition of several signalling pathways [193]. This also applies to TREM2/DAP12 signalling, which may have either pro-inflammatory or anti-inflammatory effects [194,195].

The mechanisms that would explain the involvement of other tissues remain elusive. In short, we still do not know the details of the biochemical changes that account for the complex and diverse pathology in NHD. Some enigmatic data must also be clarified. Dap12 and Trem2 are expressed from embryonic stage to adulthood; microglial cells and oligodendrocytes were identified as the major Dap12/Trem2-producing cells in the CNS, thus suggesting both microglia and oligodendrocytes as key players in PLOSL pathogenesis [196]. TREM2 has also been described in oligodendrocytes during development, but not in adulthood [173,176]. The localization and timing of the expression is important, as the abnormal priming of oligodendrocyte precursors during development by dysfunctional microglial cells or precursors may be the cause of altered function of adult oligodendrocytes. Another study showed TREM2/DAP12 expression, not only in microglia, but also in a fraction of neurons in both human and mouse cerebral cortex, and in a glioblastoma cell line; neuronal TREM2 expression was rare in the hippocampus. Astrocytes and oligodendrocytes were TREM2 negative. In neurons and microglia, the receptor appeared to be mostly located intracellularly, partially coinciding with (or adjacent to) the Golgi complex/trans-Golgi network [197]. The differences in the localization of TREM2/DAP12 probably depend on the antibodies and methods used to detect the proteins. However, studies are needed to elucidate the expression and timing of TREM2 in different cell types.

## 4. Concluding Remarks

The categorization of POLD/HDLS/ALSP and NHAD/PLOSL as primary microglial leukodystrophies or leukoencephalopathies is not completely correct. Mutations in colony stimulating factor 1 receptor (*CSF1R*) cause CSF-1R-related leukoencephalopathy (ALSP-CSFR1 or CRL, OMIM # 221820). Recessive mutations in alanyl-transfer (t)RNA synthase 2 (*AARS2*) cause leukoencephalopathy, progressive with ovarian failure (ALSP-AARS2 or LKENP, OMIM # 615889). A third, still poorly defined group of ALSP-like leukoencephalopathies is linked to mutations in *AARS1.* CRL is a microglial leukoencephalopathy whereas leukoencephalopathies linked to *AARS* mutations are not. Since not all ALSP cases have been genetically identified, we cannot ignore the possibility that mutations in other genes might be causative of ALSP variants, whether or not linked to microglia. Recent clinical reviews have focused on the diagnosis of adult leukodystrophies, and particularly on “microglial leukoencephalopathies” [198,199,200]. Hematopoietic stem cell transplantation has been used in a few clinical trials with variable results [110,199,201,202,203,204]. More specific attempts to treat CSF1R-microglial encephalopathy are focused on microglial-based therapies [205].

PLOSL or NHD is caused by mutations in two different genes, either TYROBP or TREM2, which cause slightly different versions of the disease designated as PLOSL1 and PLOSL2, respectively. Leukoencephalopathy, thalamic dysmyelination, altered synaptogenesis, and neuron loss in NHD have been interpreted as the result of microglia impairment. For this reason, PLOSL has been considered a primary microgliopathy [1,7]. Since microglia and osteoclasts are affected in NHD, redefinition of the disease as a multisystemic “immunological” disease has also been proposed [110]. Treatments directed only to microglia will be inefficient to cover the complex pathology in NHD.

## Figures and Tables

**Figure 1 ijms-23-06341-f001:**
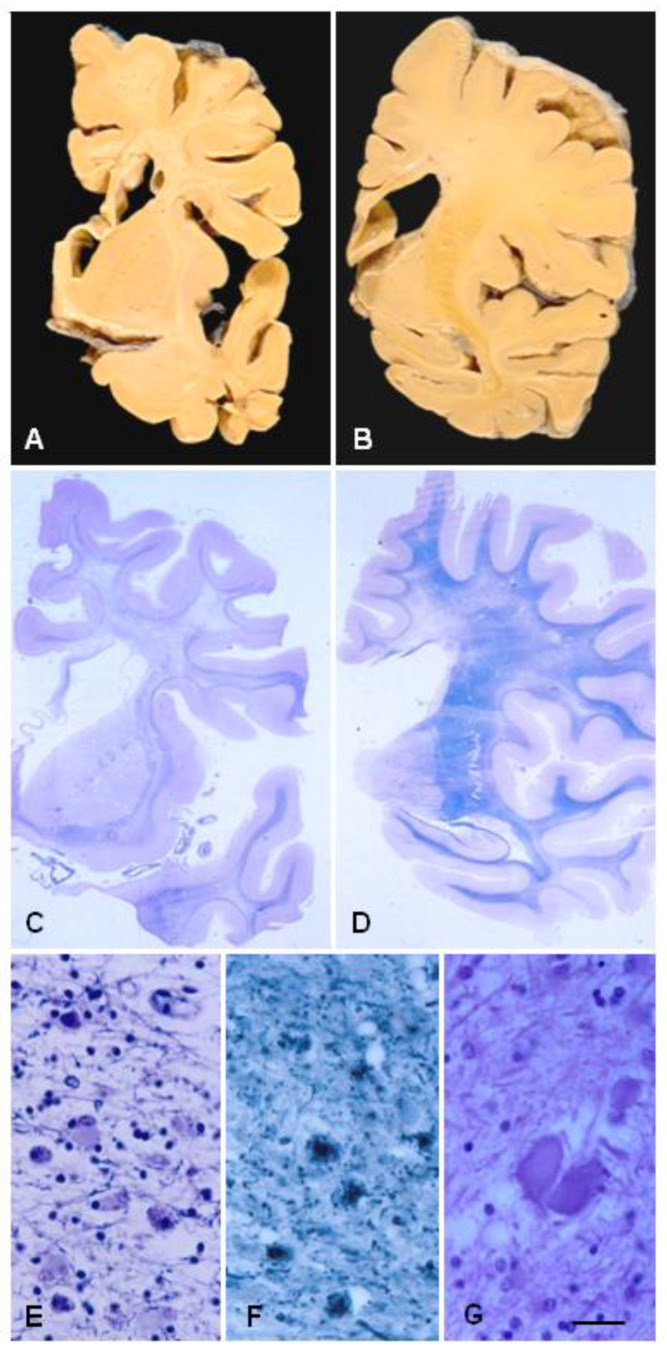
ALSP. (**A**,**B**) Coronal sections of the brain showing atrophy of the cerebral white matter more markedly in the anterior poles than in the posterior regions, atrophy of the corpus callosum, and enlargement of the lateral ventricles. (**C**,**D**) Hemispheric sections stained with Klüver–Barrera showing severe myelin loss of the centrum semi-ovale, corpus callosum, and internal capsule, and preservation of the short cortico-subcortical U-fibres. (**E**,**F**) Astrocytic gliosis and macrophages filled with pigment in the affected white matter. (**G**) Axonal ballooning. Paraffin sections, (**E**–**G**), bar = 25 μm.

**Figure 2 ijms-23-06341-f002:**
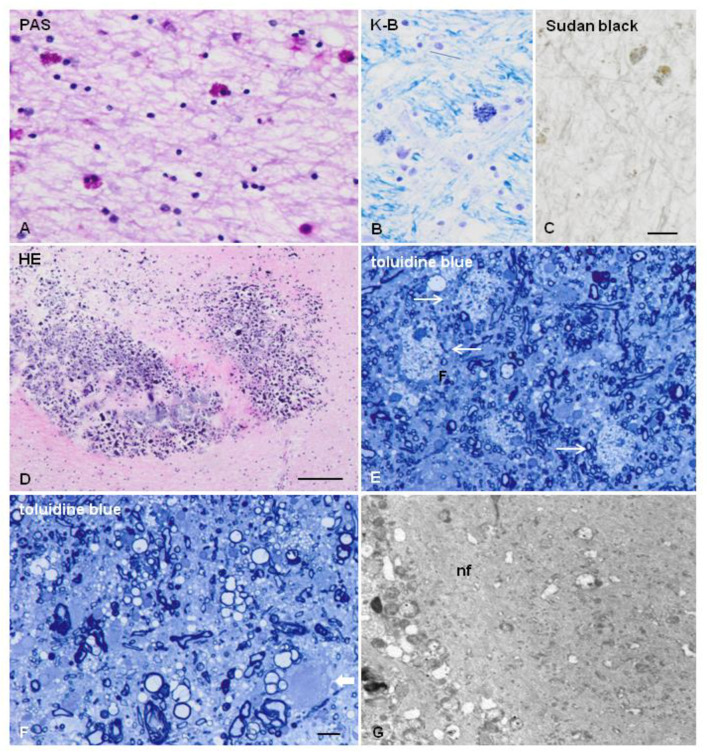
ALSP. (**A**–**C**) Glial cells filled with pigment as seen with periodic acid–Schiff (PAS) (**A**), Klüver–Barrera (K-B) (**B**), and Sudan black (**C**) staining in the frontal white matter. (**D**) Microcalcifications in the frontal white matter near the lateral ventricle. (**E**,**F**) Semi-thin sections stained with toluidine blue showing loss of myelin, macrophages filled with neutral lipids (thin arrows in (**E**)), and axonal ballooning (thick arrow in (**F**)). (**G**) Axonal ballooning filled with neurofilaments (nf) in the white matter. (**A**–**D**), formalin-fixed samples, paraffin sections; (**E**–**G**), samples fixed with glutaraldehyde, post-fixed with osmium tetraoxide, and embedded in araldite; ultra-thin sections stained with uranyl acetate and lead citrate. (**A**–**C**), bar = 25 μm; (**D**) = 100 μm; (**E**–**F**) = 10 μm.

## Data Availability

Not applicable.

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
