# Peer review of "The Primary Microglial Leukodystrophies: A Review"

_ijms, 2022, doi:10.3390/ijms23116341_

Round 1

Reviewer 1 Report

A well-written review on leukodystrophies. The review clearly articulates the clinical features, radiological findings, general and neuropathological findings, genetics and pathogenesis of leukodystrophies. The author argues that whilst mutations in the colony stimulating factor 1 receptor (CSF1R) cause CSF-1R-related leukoencephalopathy (ALSP-CSFR1 or CRL), recessive mutations in alanyl-transfer (t)RNA synthase 2 (AARS2) cause leukoencephalopathy, progressive with ovarian failure (ALSP-AARS2 or LKENP) are not microglial leukoencephalopathies.

The review also discusses NHD and the involvement of both microglia and osteoclasts, supporting the view that this is a multisystemic “immunological” disease and that treatments targeting microglia will not address the entire disease phenotype.

Reviewer 2 Report

The proposed manuscript is well structured and fine written. Although, in this overview some graphical illustrations and explanations of the genetics or patogenesis part would be very acceptable as well. 

In general, the author provide an acceptable overview of a topical clinical issue and that could certainly attract the attention of the relevant professionals.  

Reviewer 3 Report

This is a very nice, balanced review on the primary microglial leukodystrophies. It lists all the diseases that belong to the group, and several other similar disorders. It is beautifully illustrated and well written. After a spellcheck it can be accepted as is.